# Diagnostic Relevance of miR-185, miR-141, and miR-21 in Colon Carcinoma: Insights into Tumor Sidedness and Reference Gene Selection

**DOI:** 10.3390/biomedicines13102460

**Published:** 2025-10-10

**Authors:** Dorian Kršul, Ema Prenc, Lidija Požgaj, Dora Štefok, Paula Pongrac, Marija Podolski, Andrea Paravić Radičević, Damir Karlović, Ante Jerković, Marin Golčić, Ivan Dražić, Sandra Glavaš Kršul, Dora Fučkar Čupić, Vesna Eraković Haber, Marko Zelić

**Affiliations:** 1Clinical Hospital Center Rijeka, Krešimirova 42, 51000 Rijeka, Croatia; damir.karlovic@yahoo.com (D.K.); ante.jerkovic01@gmail.com (A.J.); marin.golcic@gmail.com (M.G.); sandra.glavas.krsul@gmail.com (S.G.K.); dora.fuckar.cupic@medri.uniri.hr (D.F.Č.); marko.zelic@uniri.hr (M.Z.); 2Faculty of Medicine, University of Rijeka, Braće Branchetta 20, 51000 Rijeka, Croatia; vesna.erakovichaber@selvita.com; 3Selvita Ltd., Prilaz baruna Filipovića 29, 10000 Zagreb, Croatia; ema.prenc@selvita.com (E.P.); lidija.pozgaj@selvita.com (L.P.); dora.stefok@selvita.com (D.Š.); paula.pongrac@selvita.com (P.P.); marijapodolski1@gmail.com (M.P.); andrea.paravicradicevic@selvita.com (A.P.R.); 4Faculty of Engineering, University of Rijeka, Vukovarska 58, 51000 Rijeka, Croatia; ivan.drazic@riteh.uniri.hr; 5Faculty of Biotechnology and Drug Development, University of Rijeka, Radmile Matejčić 2, 51000 Rijeka, Croatia

**Keywords:** colorectal cancer, miR-185, miR-141, miR-21, miR-16, biomarkers, FFPE tissue

## Abstract

**Background/Objectives:** MicroRNAs (miRNAs) regulate gene expression and are proposed as biomarkers in colorectal cancer (CRC). This study evaluated miR-185-5p, miR-141-5p, and miR-21-5p expression in CRC tissues; their association with tumor location, histopathology, and clinical outcomes; and the suitability of miR-16-5p and miR-151a-3p as housekeeping controls. Previous reports suggest tumor-suppressive roles for miR-185 and miR-141 and an oncogenic function for miR-21, though findings remain inconsistent. **Methods:** Paired tumor and adjacent normal tissues from 70 CRC patients were analyzed. RNA was extracted from FFPE samples, and miRNA expression quantified by RT-qPCR. Relative expression values were normalized to miR-151a-3p. Tumor–normal differences, localization effects, and associations with clinicopathological and outcome variables were assessed using repeated-measures ANOVA and non-parametric tests. **Results:** miR-185-5p and miR-141-5p were significantly reduced in tumors compared with normal mucosa while miR-21-5p was upregulated. miR-16-5p showed higher expression in normal tissue, indicating its instability and unsuitability as a housekeeping control. A modest but significant localization effect was observed for miR-185, while other miRNAs were minimally influenced by location. Baseline asymmetry between non-tumor samples, observed for miR-185-5p, further indicated sidedness effects. None of the miRNAs were associated with stage, histological type, grade, invasion, immune infiltration, progression, or five-year survival. **Conclusions:** miR-185-5p, miR-141-5p, and miR-21-5p show robust tumor–normal differences, supporting their diagnostic potential, while miR-16-5p is unsuitable as a housekeeper. Modest but significant localization effect was observed for miR-185 in right-sided tumors. None showed prognostic value in stage I–III CRC. Larger, location-stratified studies are warranted.

## 1. Introduction

Colorectal cancer (CRC) is one of the most common cancers worldwide, ranking third in incidence after lung and breast cancer, and second in mortality following lung cancer. There are 1.9 million new cases of CRC and 904,000 deaths due to CRC [1]. The incidence of CRC continues to rise in lower-income countries, while it stagnates or decreases in high-income countries, although the incidence of this malignancy is still the highest globally. Risk factors include alcohol consumption, the intake of red and processed meat, smoking, obesity and physical inactivity [2].

Colon carcinoma (CC) arises from the epithelial cells of the colonic mucosa and accounts for approximately 95% of all colon malignancies [3]. Most cases develop from precancerous polyps through the well-established adenoma-carcinoma sequence [4,5]. CRC can develop via three main genetic pathways: chromosomal instability (CIN), microsatellite instability (MSI) or mismatch repair deficiency (MMR) and the CpG island methylator phenotype (CIMP) [3,4,5,6].

Although the umbrella term “colorectal cancer” has long been used, significant differences in carcinogenesis, tumor biology, and treatment response exist between colon and rectal cancers [7,8]. In this manuscript, we will refer to our cohort as CC, reflecting the origin of the analyzed samples. However, when citing previous studies, we will use the term CRC, as this was the terminology employed in the referenced publications.

Similarly, substantial molecular, genetic, and histological differences are observed between right-sided and left-sided colon cancers. Left-sided tumors often exhibit a polypoid appearance and are typically tubular, villous, or conventional adenocarcinoma. In contrast, right-sided tumors tend to be flat and more frequently originate from sessile serrate adenomas, often showing mucinous features [9]. Differences in molecular profiles between left-sided and right-sided CRC are well documented, including variations in gene mutations, pathways, and even some microRNA expression patterns [10]. Additionally, the gut microbiota composition differs markedly between tumor sites, potentially contributing to tumor development and progression [11].

Dysregulated miRNA expression, either upregulation or downregulation, contributes to tumor development by modulating key cancer-related pathways, with individual miRNAs acting as oncogenes (oncomiRs) or tumor suppressors depending on the cellular context [12]. Previous studies have identified miRNAs as a promising biomarker for the diagnosis and treatment of CRC [13] and proposed tumor suppressor roles for miR-185 and miR-141, and oncogenic function for miR-21 in CRC. However, findings remain inconsistent across literature, likely due to differences in study design, sample types and analytical approaches. Accurate miRNA expression analysis in CRC depends on robust normalization using stably expressed reference miRNAs that are not differentially expressed between sample groups. However, no single “universal” reference miRNA works for all tissues or experimental conditions. Therefore, identifying appropriate housekeeping miRNAs—also called reference or normalizer miRNAs—is crucial for quantitative RT-PCR (qRT-PCR) and related assays [14].

Earlier studies in CRC tissue identified a panel of miRNAs with minimal expression differences between tumor and normal colonic mucosa. For the purposes of this research miR-16-5p and miR-151-3p were used as reference miRNAs which have been proven to be stably expressed “housekeeping” miRNAs in CRC studies [15,16].

The purpose of this study was to investigate whether the expression of miR-185-5p, miR-141-5p, and miR-21-5p differs between CC tissue and adjacent non-tumorous mucosa, and whether these differences vary according to tumor location in the left versus right colon. By analyzing paired tissue samples from a well-defined cohort of surgically resected CC patients, we sought to clarify the potential of these miRNAs to serve as diagnostic or prognostic biomarkers, and to determine if their expression reflects the known biological heterogeneity between proximal and distal colon cancers. In addition, we assessed the stability of miR-16-5p and miR-151a-3p as endogenous controls to evaluate their suitability as housekeeping miRNAs for normalization of candidate miRNA in CC studies.

## 2. Materials and Methods

### 2.1. Study Population

This study included tissue samples from 70 patients who underwent colon resection at the Department of Surgery, University Hospital Centre Rijeka, starting in January 2019. For each patient, paired samples of tumor tissue and adjacent non-tumorous (healthy) tissue were collected, resulting in a total of 140 samples.

Samples were collected consecutively, in the order in which patients underwent surgery. A cohort consisted of 41 male and 29 female patients, 44 to 88 years old, that had histologically confirmed colon adenocarcinoma. Only patients whose treatment was initiated with curative intent were included. At the time of diagnosis, the disease was localized, corresponding to pathological stages I through III.

Patients were divided into two equal groups based on tumor location: 35 patients with left-sided CC and 35 with right-sided CC.

Exclusion criteria included: patients who underwent palliative resections, had distant metastases at diagnosis, were undergoing active treatment for or had a history of another primary malignancy, or died within 90 days postoperatively. To avoid phenotypic overlaps between right- and left-sided disease, patients with tumors located in the transverse colon were excluded. Patients with rectal cancer were also not included in the study.

### 2.2. Sample Collection and RNA Analysis

Tissue samples were preserved as formalin-fixed, paraffin-embedded (FFPE) blocks. Total microRNA was isolated from both tumor tissue and adjacent non-tumorous tissue. The expression levels of selected microRNAs were then quantified. For each patient, the adjacent healthy tissue was used as a matching control for comparative analysis.

### 2.3. Data Stratification and Follow-Up

Patients were stratified by tumor location (left-sided vs. right-sided), tumor stage at the time of surgery, disease progression status, and overall survival.

All patients were followed postoperatively for a period of up to five years or until death, whichever occurred first.

### 2.4. RNA Isolation

Total RNA (including miRNA) was isolated from two 10 µm thick sections of formalin-fixed paraffin-embedded colon tissue blocks. Deparaffinization of sections was performed using xylene, followed by total RNA isolation using miRNeasy FFPE Kit (Qiagen, Hilden, Germany) according to manufacturer’s instructions. Total RNA was eluted in 20 µL of RNase free H_2_O and concentration was determined by using Quant iT RiboGreen RNA assay kit (Invitrogen, Waltham, MA, USA).

### 2.5. Reverse Transcription (RT) and qPCR

MiRNA was reverse transcribed into cDNA using miRCURY LNA RT kit (Qiagen) following manufacturer’s instructions. Each RT reaction contained 140 ng of total RNA in a final volume of 10 μL and was incubated for 60 min at 42 °C, 5 min at 95 °C and then held at 4 °C in Mastercycler X50s (Eppendorf, Hamburg, Germany). Relative expressions of miR-185-5p, miR-141-5p and miR-21-5p were determined by RT-qPCR method using miR-151a-3p and miR-16-5p as endogenous controls. For detection of miRNAs, miRCURY LNA SYBR Green PCR kit (Qiagen) with miRCURY LNA miRNA PCR Assays specific to each miRNA were used (YP00205702, YP00204576, YP00206088, YP00204230, YP00206037). Reactions were performed as follows: 2 min at 95 °C (1 cycle) and 40 interchangeable cycles of 10 sec at 95 °C and 60 sec at 56 °C in Quant Studio 6 (Applied Biosystems, Waltham, Massachusetts, USA).

### 2.6. Data Analysis

Total RNA concentration in samples was calculated by interpolation from RNA standard curve using Microsoft Excel.

Mean Ct value of endogenous control (miR-151a-3p) was subtracted from mean values for test genes (miR-185-5p, miR-141-5p, miR-21-5p, miR-16-5p) and relative gene expression was further calculated from ΔCt using equation 2^−ΔCt^. Although both miR-16-5p and miR-151a-3p were initially tested, miR-16-5p showed significant tumor–normal differences and was therefore excluded from normalization, leaving miR-151a-3p as the sole reference.

To assess stability, we compared Ct distributions of candidate references across tissue type and tumor localization, with the assumption that an ideal reference shows minimal variation between groups.

Statistical analyses were performed using JASP (version 0.17.2.1; JASP Team, University of Amsterdam, Amsterdam, The Netherlands). To assess the distribution of the data, we applied the Shapiro–Wilk test of normality, which is the default and recommended procedure in JASP. Alongside *p* values, ω^2^ was reported as a measure of effect size. This was performed because a result can be statistically significant yet represent only a trivial difference with little practical relevance. By quantifying the proportion of variance explained by the factor, ω^2^ addresses this limitation and provides a more informative indication of the substantive importance of group differences. For non-normally distributed data, the Mann–Whitney U test with rank-biserial correlation was applied, while one-way ANOVA was used for comparisons across multiple groups. Associations with continuous clinicopathological variables were examined using Pearson’s correlation. Statistical significance was set at *p* < 0.05.

## 3. Results

### 3.1. Evaluation of Candidate Housekeeping miRNAs

To assess the stability of the endogenous controls, expression levels of miR-151a-3p and miR-16-5p were compared between tumor and adjacent non-tumorous tissue, as well as across left- and right-sided colon cancers.

For miR-151a-3p, repeated measures ANOVA demonstrated a statistically significant difference between tumor and normal tissue (*p* = 0.012). However, the effect size was small (η^2^ = 0.050), indicating limited biological impact. No significant interaction with tumor side was observed (*p* = 0.776), and between-subjects analysis confirmed no difference between left- and right-sided tumors (*p* = 0.455). Descriptive statistics likewise showed highly similar mean Ct values across groups, supporting the overall stability of miR-151a-3p as a reference gene.

In contrast, miR-16-5p showed a more pronounced difference between tumor and normal tissues (*p* < 0.001), with an effect size in the medium-to-large range (η^2^ = 0.093). Although no significant side interaction was detected (*p* = 0.498), this finding indicates that miR-16-5p expression is systematically altered by disease state, compromising its suitability as a neutral housekeeping miRNA.

Taken together, these results support miR-151a-3p as the more robust endogenous control for normalization in this dataset, whereas miR-16-5p should be used with caution due to its disease-associated variability.

### 3.2. MiRNA Expression Analysis

Relative expression values, normalized against miR-151a-3p, were used for all comparisons.

Two-way repeated-measures ANOVA was performed to compare miRNA expression between tumor and adjacent normal mucosa, stratified by tumor sidedness (right vs. left colon). Across all four miRNAs, tissue type emerged as the dominant factor, whereas the effect of tumor localization was smaller, and interactions were generally absent or minor. (Table 1.)

For miR-185-5p, expression was significantly higher in non-tumor mucosa compared with tumor tissue (*p* < 0.001; ω^2^ = 0.082). Localization also showed an independent effect, with modestly higher levels in right-sided samples (*p* < 0.001; ω^2^ = 0.104). In addition, the size of the tumor–non-tumor difference was not the same in the right and left colon (*p* = 0.010; ω^2^ = 0.038), meaning that the effect of tumor status partly depended on the side of the colon from which the sample was taken.

For miR-141-5p, expression was also higher in normal tissue compared to tumor (*p* < 0.001; ω^2^ = 0.094). Localization exerted a smaller but statistically significant effect (*p* = 0.027; ω^2^ = 0.030), while no significant interaction was found (*p* = 0.930). Thus, the tumor–normal difference was consistent across both colon sides.

In contrast, miR-21-5p showed the opposite pattern, with significantly higher expression in tumor compared to normal mucosa (*p* < 0.001; ω^2^ = 0.094). Neither localization (*p* = 0.094) nor interaction (*p* = 0.753) reached significance, indicating that this upregulation was consistent regardless of tumor side.

The strongest effect was observed for miR-16, which displayed markedly higher expression in normal mucosa compared with tumor (*p* < 0.001; ω^2^ = 0.190). Localization also contributed to a smaller effect (*p* = 0.012; ω^2^ = 0.041), while no significant interaction was detected (*p* = 0.403). (Table 1.)

Overall, these findings confirm that tissue type (tumor vs. normal) is the main determinant of miRNA expression, with miR-21-5p (upregulated in tumors) and miR-16-5p (downregulated in tumors) showing the most robust and consistent differences. Localization played a secondary role, most evident for miR-185-5p, whereas interaction effects were generally minimal. Interaction plots for each miR are presented in Figure 1.

### 3.3. Association of miRNA Expression with Clinicopathological Features

We next examined the relationship between miR-185-5p, miR-141-5p, miR-21-5p, and miR-16-5p expression in tumor tissue and standard pathological parameters, including T stage, N stage, histological type, tumor grade, lymphovascular invasion, lymphocytic infiltration, and perineural invasion.

#### 3.3.1. Tumor Stage (T and N)

Relative expression of all four miRNAs did not differ significantly between T1–2, T3, and T4 tumors (*p* > 0.5 for all comparisons). Similarly, no significant variation was observed across nodal categories N0, N1, and N2. Although descriptive analyses suggested slightly higher miR-185-5p and miR-21-5p levels in node-positive tumors, these differences were not statistically significant (*p* = 0.126 and *p* = 0.367, respectively). (Table 2.)

Values are presented as mean ± standard deviation. Differences between groups were analyzed using ANOVA. Effect sizes are reported as ω^2^. No statistically significant associations were observed between miRNA expression and tumor stage.

#### 3.3.2. Histological Subtype and Grade

Comparison between adenocarcinoma and mucinous carcinoma revealed no significant differences in expression for any of the analyzed miRNAs (all *p* > 0.3). Likewise, expression patterns did not vary between high- and low-grade tumors (*p* > 0.5 for all), indicating no association with tumor differentiation. (Table 3.)

Values are presented as median (range). Differences between groups were assessed using the Mann–Whitney U test. Effect sizes are reported as rank-biserial correlation.

#### 3.3.3. Lymphovascular Invasion and Lymphocytic Infiltration

The presence of lymphovascular invasion did not significantly influence miRNA levels. Median expression values of miR-185-5p, miR-141-5p, miR-21-5p, and miR-16-5p were comparable between tumors with and without invasion (all *p* > 0.28). Similarly, no significant differences were observed among tumors with abundant, moderate, or sparse lymphocytic infiltration (all *p* > 0.23), although variability was noted within subgroups. (Table 4.)

#### 3.3.4. Perineural Invasion

Due to the small number of cases with perineural invasion (*n* = 4), statistical testing was not feasible.

Values are presented as mean ± standard deviation or median (range), depending on data distribution. Differences between groups were tested by Mann–Whitney U test (for lymphovascular invasion) or ANOVA (for lymphocytic infiltration). Effect sizes are reported as rank-biserial correlation or ω^2^, respectively. Analysis for perineural invasion was not performed due to insufficient sample size.

#### 3.3.5. Tumor Size

Finally, correlation analysis demonstrated no significant association between tumor size and miRNA expression for any of the four candidates (Pearson’s r: −0.016 for miR-185, −0.010 for miR-141, 0.106 for miR-21, and 0.018 for miR-16; all *p* > 0.38).

Overall, these analyses demonstrate that miR-185, miR-141-5p, miR-21-5p, and miR-16-5p expression in tumor tissue is not significantly associated with key histopathological parameters, suggesting limited diagnostic or prognostic utility in this context.

### 3.4. Association of miRNA Expression with Clinical Outcomes

We next evaluated whether tumor expression of miR-185-5p, miR-141-5p, miR-21-5p, and miR-16-5p was associated with clinical outcomes, including disease progression and 5-year survival.

For disease progression, there were no significant differences in relative expression between patients who experienced progression (*n* = 9) and those without progression (*n* = 59). Median expression values were similar across groups (miR-185-5p: 1.158 vs. 0.800, *p* = 0.479; miR-141-5p: 0.055 vs. 0.038, *p* = 0.949; miR-21-5p: 207.5 vs. 155.9, *p* = 0.978; miR-16-5p: 9.98 vs. 6.87, *p* = 0.978). Effect sizes for all comparisons were small (|r| < 0.15), indicating negligible prognostic impact.

Similarly, for 5-year survival, no significant differences in tumor expression of the four miRNAs were observed between survivors (*n* = 54) and non-survivors (*n* = 15). Median levels remained comparable (miR-185-5p: 0.831 vs. 0.764, *p* = 0.871; miR-141-5p: 0.036 vs. 0.052, *p* = 0.690; miR-21-5p: 149.5 vs. 184.6, *p* = 0.451; miR-16-5p: 6.83 vs. 8.89, *p* = 0.657), with small effect sizes (|r| ≤ 0.13). (Table 5.)

These results indicate that none of the examined miRNAs were associated with disease progression, long-term survival, or tumor size, suggesting a lack of prognostic value in this cohort.

### 3.5. Correlation Analysis of miRNA Expression

Pairwise correlation analysis demonstrated strong and consistent positive associations among the miRNAs investigated in both tumor and normal tissues. Associations with continuous clinicopathological variables, as well as correlations between different miRNAs, were examined using Spearman correlation.

In tumor tissue, the strongest correlations were observed between miR-16-5p and miR-141-5p (r ≈ 0.92), miR—21-5p and miR-141-5p (r ≈ 0.90) and between miR-16-5p and miR-21-5p (r ≈ 0.95). (Figure 2.) Moderate positive associations were also present between miR-185-5p and the other miRNAs (r ≈ 0.62–0.67). These findings indicate that miR-16, miR-141, and miR-21 form a tightly interconnected expression cluster in malignant tissue, with miR-185 showing weaker integration. (Figure 2.)

In normal mucosa correlations were strong, particularly between miR-16-5p and miR-141-5p (r ≈ 0.89) and between miR-21-5p and miR-141-5p (r ≈ 0.84). MiR-185 again displayed weaker associations with the other miRNAs (r ≈ 0.67–0.75). This suggests that in healthy colonic mucosa, expression of miR-16, miR-141, and miR-21 is tightly co-regulated, while miR-185 is only partially coordinated.

These results indicate that miR-16, miR-141, and miR-21 are strongly co-expressed in both tumor and non-tumor tissues, suggesting overlapping regulatory control. MiR-185 shows weaker but consistent correlations, pointing to a more context-specific role in CRC biology.

## 4. Discussion

In this study, we evaluated the expression of miR-185-5p, miR-141-5p, miR-21-5p, and miR-16-5p in CC tissue, compared with adjacent normal mucosa, and explored their associations with tumor localization, histopathological features, and clinical outcomes. Our findings provide insight into the diagnostic and prognostic potential of these candidate biomarkers.

Initially, both miR-16-5p and miR-151a-3p were selected as candidate housekeeping miRNAs for normalization. However, statistical analysis revealed that miR-16-5p exhibited significant variability between tumor and normal tissues, indicating that it was not a stable reference in this context.

In addition to our internal findings, emerging literature challenges the suitability of miR-16-5p as a reference miRNA in CRC. The experimental study by Huang et al. (2022) demonstrated that miR-16-5p is significantly downregulated in CRC tissues compared with adjacent normal mucosa, and that it acts as a potent tumor suppressor by targeting FOXK1 and inhibiting the PI3K/Akt/mTOR pathway [17]. Overexpression of miR-16-5p suppressed proliferation, angiogenesis, and enhanced apoptosis in CRC cell lines, while its inhibition produced the opposite effect. Similarly, the comprehensive review by Ghafouri-Fard et al. (2022) concluded that miR-16-5p downregulation is a nearly universal finding across solid and hematologic malignancies, including CRC, with its loss linked to poor differentiation, higher metastatic potential, and worse clinical outcomes [18].

This data indicates that miR-16-5p is not a neutral housekeeping transcript but instead a biologically active tumor suppressor miRNA with consistent dysregulation in cancer. In contrast, miR-151a-3p showed only minimal variation across tissues and no association with tumor side, confirming it as the more robust internal control in our dataset. For these reasons, miR-151a-3p should be preferred over miR-16-5p as a housekeeping miRNA in CC tissue studies, while miR-16-5p should be avoided due to its consistent involvement in tumor biology and potential to bias expression analyses.

MiRNA-185-5p is frequently dysregulated in CRC, with most studies reporting its downregulation in tumor tissues compared to normal colon mucosa. Tumor specimens often exhibit significantly lower miR-185-5p levels than adjacent normal tissue, and this reduction correlates with advanced disease stages such as presence of liver metastasis, and higher likelihood of relapse [19]. Restoring miR-185-5p expression in CRC cell lines has been shown to inhibit malignant phenotypes, whereas its loss enhances tumorigenic traits, supporting its role as a tumor-suppressor miRNA in CRC [19,20]. While most evidence supports its tumor-suppressive role, some studies have reported elevated miR-185-5p expression in CRC, suggesting that its function may be context-dependent. In certain tumor settings, miR-185-5p may exhibit oncogenic properties [21]. Thus, although miR-185-5p predominantly acts as a tumor suppressor in CRC, tumor heterogeneity may allow it to acquire pro-tumorigenic roles under specific conditions.

Our data demonstrates that miR-185-5p was significantly reduced in right-sided tumor tissue compared with adjacent normal mucosa, consistent with its characterization as a tumor-suppressive miRNA. The difference was markedly bigger than in left-sided tumors. MiR-185 directly targets several oncogenic pathways in colon cancer cells, and its downregulation facilitates tumor progression. These pathways include the Wnt/beta-catenin signaling, cytoskeletal and invasion regulators, such as Rho GTPases and AQP5, and growth factor signaling pathways including IGF axis. Through these mechanisms, miR-185 can induce cell cycle arrest, promote apoptosis, and inhibit epithelial–mesenchymal transition (EMT) in CRC models [20,22,23,24,25,26].

The elevated expression of miR-185 in non-tumor tissue of the right colon, compared to the left, suggests that its regulation is primarily influenced by regional physiological factors rather than tumor-specific changes. This may reflect the unique microenvironmental characteristics of the proximal colon, including its distinct microbiota composition, immune cell landscape, and exposure to luminal metabolites such as bile acids and short-chain fatty acids. These elements could contribute to the upregulation of miR-185 as part of a homeostatic or protective regulatory mechanism. Given that miR-185 targets key components of the Wnt/β-catenin and EMT signaling pathways, its higher expression in normal right-sided colon tissue may indicate a baseline suppression of these oncogenic pathways, potentially linked to regional differences in epithelial turnover, differentiation, or immune surveillance. Furthermore, if tumors develop in a context where miR-185 is constitutively elevated, they may need to downregulate miR-185 to enable activation of Wnt/EMT signaling required for progression. This dynamic could contribute to the distinct molecular and clinical behavior of right-sided colorectal cancers, which are frequently more aggressive and biologically divergent from their left-sided tumors. Although the effect size was limited, such location-dependent variation suggests that the diagnostic utility of miR-185 could be further refined by accounting for tumor sidedness.

MiR-141-5p is a member of the miR-200 family of microRNAs, which includes miR-200a, miR-200b, miR-200c and miR-429. This family is organized into two genomic clusters, with miR-141-5p and miR-200a located on chromosome 12 [27]. The miR-200 family is well established as a key regulator of EMT, primarily through shared seed sequences that target critical EMT-inducing transcription factors, notably ZEB1 and ZEB2 [28,29]. By downregulating these transcription factors, miR-200 family promotes the transcription of E-cadherin, thereby maintaining the epithelial phenotype. This regulatory axis is considered a principal mechanism through which the miR-200 family, including miR-141-5p, exerts its tumor-suppressive effects in CRC [30]. Accordingly, miR-141-5p plays a critical role in inhibiting EMT and the initiation of metastasis.

Expression analyses of miR-141 in CRC have yielded mixed results, suggesting context-dependent role. Several studies, often conducted in cell lines and tumor tissues, report downregulation of miR-141 in primary CRC compared to normal colonic tissue. For example, Liang et al. found significantly lower levels of miR-141-3p in CRC tissues and cell lines, and patients with low tumor expression of miR-141-3p had worse outcomes [27]. Conversely, other studies, particularly those involving patient serum analyses or advanced-stage tumors, report upregulation of miR-141 in later stages or metastatic CRC. Li et al., for instance, observed an increase in miR-141 expression with advancing tumor stage [31]. These findings suggest a complex expression pattern: it may be suppressed during early tumorigenesis, enabling EMT and proliferation but later becomes upregulated in certain subtypes or metastatic settings.

In this study, miR-141-5p expression was lower in tumor tissue relative to normal mucosa, reinforcing its role as part of the miR-200 family in maintaining epithelial phenotype and preventing EMT. These results support miR-141-5p as a marker of malignant transformation, even though no association was found with histological type, grade, invasion, or clinical outcomes in this cohort. Unlike studies that have reported higher miR-141, our cohort did not include patients with advanced or metastatic CRC.

MiR-21-5p is a short noncoding RNA that has emerged as a prototypical “oncomiR”, a microRNA with oncogenic effects. It was one of the first microRNAs found to be upregulated across multiple human cancers, including CC [32]. In the context of CC, miR-21-5p dysregulation is linked to advanced disease stage and aggressive tumor behavior [33]. MiR-21-5p is significantly overexpressed in CC tissues compared to normal colonic mucosa, with elevated levels observed even from early stages and tending to increase in higher tumor stages [32,33]. This overexpression was found to be particularly prominent in fibroblast-like tumor stromal cells, reflecting a tumor promoting microenvironment [34].

MiR-21-5p functions as a post-transcriptional repressor of multiple tumor suppressor genes, thereby promoting oncogenic pathways in colon cancer. Elevated miR-21-5p levels are known to repress tumor suppressors such as PTEN and PDCD4, thereby promoting proliferation, invasion, and survival [32,35,36].

In this study, miR-21-5p was clearly upregulated in tumor tissue, confirming its role as a canonical oncomiR in CRC. Its tumor–normal difference was robust and consistent across tumor locations, underscoring its general involvement in colorectal carcinogenesis. Most studies report that miR-21-5p is upregulated in colorectal tumors regardless of location, without significant difference in overall miR-21-5p levels between left and right colon cancers. A study by Knudsen et al. examined miR-21-5p in situ in colon adenocarcinomas and noted that left-sided CC were more likely to show miR-21-5p expression in the tumor cells themselves, whereas in many right-sided tumors miR-21-5p was confined to the stromal compartment [37]. Despite this observation, there is no strong evidence that miR-21-5p behaves differently at a mechanistic level between right vs. left colon tumors.

We have not observed miR-21-5p association with tumor stage, histopathological variables, disease progression, or survival. This suggests that while miR-21-5p may reliably indicate the presence of malignancy, it does not provide information in localized CRC.

The strongest tissue-type effect in our study was observed for miR-16-5p, which was markedly higher in normal mucosa compared with tumors. MiR-16-5p has long been considered biologically neutral, yet its expression pattern in our study supports a potential role in tumorigenesis that warrants further detailed investigation.

An additional finding of our study was the presence of baseline differences in miRNA expression between left- and right-sided normal colonic mucosa. Specifically, miR-185-5p, miR-141-5p, and miR-16-5p each showed modest but statistically significant localization effects, with higher levels detected in right-sided samples, whereas miR-21-5p expression did not differ by location. These results suggest that the colonic mucosa is not biologically uniform, and that inherent differences in microRNA expression between proximal and distal colon may reflect their distinct embryological origins and molecular environments. Such baseline asymmetry, that might go beyond miRNAs explored in this study, is important to consider when interpreting tumor–normal comparisons and may partially account for side-specific patterns of tumorigenesis observed in CRC.

Our results show that miR-185-5p, miR-141-5p, and miR-21-5p demonstrate significant tumor–normal differences, supporting their use as diagnostic biomarkers in CRC. MiR-185-5p additionally reflects modest but significant differences between right- and left-sided cancers, highlighting the need to consider tumor location in biomarker studies. Furthermore, miR-16-5p, with its strong tumor–normal difference, is unsuitable as a reference gene. Importantly, none of the analyzed miRNAs correlated with histopathology or survival, underlining their limited prognostic value in localized disease. This contrasts with previous reports that suggested prognostic roles for miR-21, miR-185, and miR-141, with elevated miR-21 and reduced miR-185/miR-141 often linked to more aggressive disease and poorer survival [27,38,39]. Differences in cohort size, stage distribution, methodology, and normalization strategies may explain these discrepancies. Our negative findings suggest that while these miRNAs are dysregulated in CC, they may not independently capture the heterogeneity of tumor aggressiveness in localized disease.

The correlation analysis provides additional insight into the regulatory relationships among the studied miRNAs. The consistently strong co-expression of miR-16, miR-141, and miR-21 across both tumor and normal tissues suggests that these miRNAs may be under shared transcriptional or epigenetic control, or that they converge on common signaling pathways relevant to colorectal epithelial homeostasis and tumorigenesis. Their tight clustering also supports the idea that these molecules are not acting in isolation but may represent components of a coordinated regulatory network. In contrast, miR-185 demonstrated weaker but stable correlations with the other miRNAs, which may reflect a more specialized or context-dependent function in CRC. This relative independence could explain why miR-185 shows distinct associations with tumor side and variable behavior across published studies. Together, these findings imply that while certain miRNAs in CRC are co-regulated as part of broader oncogenic processes, others may play more targeted roles whose contribution depends on tumor biology and anatomical context.

While confirming previous reports of dysregulation, our study uniquely demonstrates that tumor–normal differences persist across locations, and that diagnostic performance may be refined by accounting for baseline asymmetry between right- and left-sided mucosa. This underscores the importance of location-stratified biomarker validation before clinical application.

This study has several limitations. The cohort size was modest and the cohort was relatively homogeneous, with limited variability in tumor stage, nodal status, and other clinicopathological parameters. The absence of advanced or metastatic cases may explain the lack of association between miRNA expression and survival, as prognostic value is more often reported in stage IV disease or mixed cohorts. Apart from tumor location, few clinical variables showed substantial diversity, which may have reduced the ability to detect meaningful associations with miRNA expression.

## 5. Conclusions

In this study, we demonstrated that miR-185-5p, miR-141-5p, and miR-21-5p are significantly dysregulated in CC tissue compared with adjacent mucosa, confirming their potential role as diagnostic biomarkers. MiR-185-5p and miR-141-5p were reduced in tumors, consistent with their tumor-suppressive functions, while miR-21-5p was upregulated, in line with its well-established oncogenic role. A modest but significant localization effect was observed for miR-185-5p, suggesting that tumor location may influence miRNA profiles. Furthermore, baseline asymmetry observed between non-tumor tissue samples suggests that tumor location is an important factor to consider when interpreting tumor–normal comparisons. MiR-16-5p showed the strongest tumor–normal difference and was unsuitable as a reference gene. None of the analyzed miRNAs correlated with stage, histopathology, disease progression, or survival, indicating that their value lies primarily in diagnosis rather than prognosis of the localized tumors. These findings highlight the diagnostic promise of tissue-based miRNA profiling in CRC and underscore the need for validation in larger, location-stratified cohorts before clinical application.

## Figures and Tables

**Figure 1 biomedicines-13-02460-f001:**
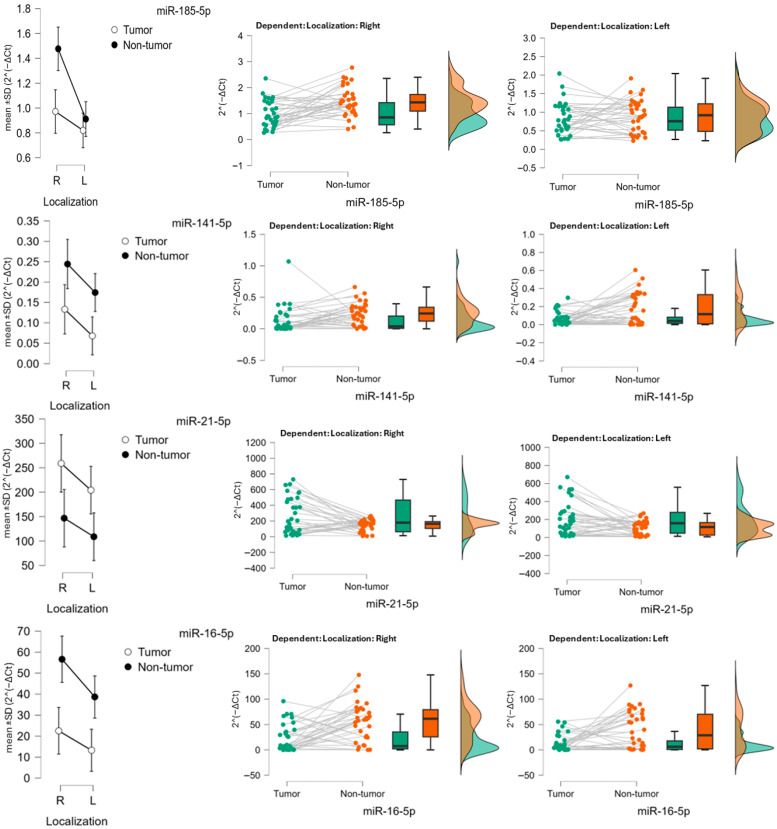
Expression of miR-185-5p, miR-141-5p, miR-21-5p and miR-16-5p in colon carcinoma. Descriptive (mean ± SD) and raincloud plots of relative expression (2^−ΔCt^) in tumor and adjacent non-tumor tissue, stratified by tumor localization (right vs. left colon).

**Figure 2 biomedicines-13-02460-f002:**
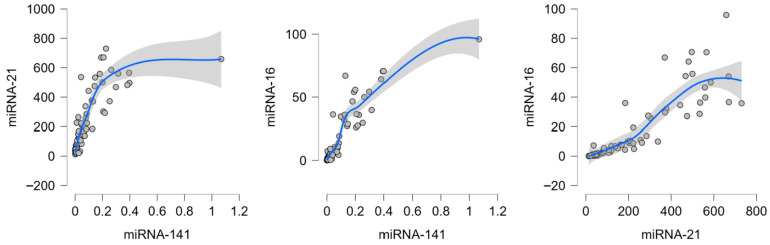
Strongest correlations between candidate miRNAs in colorectal cancer tumor tissue. Dots represent individual patient samples; the blue line indicates the LOESS regression fit, and the shaded area represents the 95% confidence interval.

**Table 1 biomedicines-13-02460-t001:** Relative expression of miR-185, miR-141, miR-21, and miR-16 in colorectal cancer and adjacent normal tissue stratified by tumor localization.

Expression	Right Localization		Left Localization		*p* * (ω^2^) (Localization)	*p* * (ω^2^) (Tissue Type)	*p* * (ω^2^) (Interaction)
	Tumor	Non-Tumor	Tumor	Non-Tumor			
miR-185-5p	0.972 ± 0.509	1.477 ± 0.053	0.820 ± 0.428	0.911 ± 0.437	<0.001 (0.104)	<0.001 (0.082)	0.010 (0.038)
miR-141-5p	0.133 ± 0.209	0.244 ± 0.170	0.068 ± 0.075	0.174 ± 0.175	0.027 (0.030)	<0.001 (0.094)	0.930 (<0.001)
miR-21-5p	259.048 ± 230.018	146.862 ± 70.829	204.287 ± 185.379	108.899 ± 78.591	0.094 (0.014)	<0.001 (0.094)	0.753 (<0.001)
miR-16-5p	22.521 ± 26.793	56.610 ± 36.948	13.279 ± 16.536	38.639 ± 36.771	0.012 (0.041)	<0.001 (0.190)	0.403 (<0.001)

* Repeated measures ANOVA. Values are presented as mean ± standard deviation.

**Table 2 biomedicines-13-02460-t002:** Relative expression of miR-185-5p, miR-141-5p, miR-21-5p, and miR-16-5p in colon cancer tissue according to tumor stage (T and N classification).

Characteristics	N (%)	Relative Expression in Tumor Tissue (Mean ± SD)			
		miR-185-5p	miR-141-5p	miR-21-5p	miR-16-5p
T stage					
T1, T2	5 (7%)	0.857 ± 0.258	0.062 ± 0.082	125.180 ± 116.361	7.927 ± 10.445
T3	58 (83%)	0.923 ± 0.487	0.103 ± 0.166	237.725 ± 213.413	18.579 ± 22.998
T4	7 (10%)	0.711 ± 0.441	0.096 ± 0.140	227.491 ± 228.848	16.942 ± 25.495
*p* *		0.529	0.859	0.521	0.602
Effect size **		<0.0001	<0.0001	<0.0001	<0.0001
N stage					
N0	54 (77%)	0.850 ± 0.459	0.099 ± 0.171	209.331 ± 214.049	15.884 ± 21.660
N1	11 (16%)	0.940 ± 0.389	0.103 ± 0.127	280.451 ± 179.025	24.738 ± 27.098
N2	5 (7%)	1.291 ± 0.637	0.095 ± 0.079	317.833 ± 215.123	20.656 ± 22.229
*p* *		0.126	0.994	0.367	0.478
Effect size **		0.032	<0.0001	<0.0001	<0.0001

* ANOVA, ** ω^2^.

**Table 3 biomedicines-13-02460-t003:** Relative expressions of miR-185-5p, miR-141-5p, miR-21-5p, and miR-16-5p according to histological type and tumor grade.

Characteristics	N (%)	Relative Expression in Tumor Tissue (Median, Range)			
		miR-185-5p	miR-141-5p	miR-21-5p	miR-16-5p
Histological type					
AC	50 (72%)	0.789 (0.264–2.354)	0.043 (0.0002–0.394)	168.455 (13.594–670.301)	7.346 (0.028–70.516)
Micropapillary AC *	1 (1%)	-	-	-	-
Mucinous AC	19 (27%)	0.831 (0.334–1.630)	0.036 (0.001–1.069)	113.477 (17.682–729.997)	4.307 (0.353–95.904)
*p* **		0. 335	0.968	1.000	0.818
Effect size ***		−0.154	−0.008	−0.001	0.038
Histological grade					
High	7 (10%)	0.874 (0.358–1.628)	0.047 (0.003–0.397)	87.236 (31.100–533.130)	5.637 (1.025–70.640)
Low	63 (90%)	0.785 (0.264–2.354)	0.038 (0.0002–1.069)	175.849 (13.594–729.997)	7.128 (0.028–95.904)
*p* **		0.531	0.788	0.976	0.835
Effect size ***		−0.147	−0.065	0.009	−0.051

* Groups with N < 5 were excluded from analysis; ** Mann–Whitney test; *** Rank-biserial correlation.

**Table 4 biomedicines-13-02460-t004:** Relative expressions of miR-185-5p, miR-141-5p, miR-21-5p, and miR-16-5p in relation to lymphovascular invasion, lymphocytic infiltration, and perineural invasion.

Characteristics	N (%)	Relative Expression in Tumor Tissue (Median, Range/Mean ± SD)			
		miR-185-5p	miR-141-5p	miR-21-5p	miR-16-5p
Lymphovascular invasion					
Yes	54 (77%)	0.764 (0.264–2.354)	0.031 (0.0002–1.069)	162.325 (13.594–729.997)	6.910 (0.028–95.904)
No	16 (23%)	0.903 (0.381–1.689)	0.057 (0.0004–0.397)	185.961 (15.296–670.301)	7.810 (0.053–70.640)
*p* **		0.289	0.175	0.352	0.509
Effect size ***		−0.177	−0.226	−0.156	−0.111
Lymphocytic infiltration					
Abundant	22 (31%)	1.040 ± 0.532	0.079 ± 0.104	235.873 ± 225.347	16.076 ± 19.781
Scarce	21 (30%)	0.855 ± 0.395	0.140 ± 0.239	238.351 ± 221.998	21.789 ± 28.446
Medium	27 (39%)	0.817 ± 0.464	0.083 ± 0.107	215.185 ± 192.943	15.633 ± 19.561
*p* +		0.237	0.364	0.915	0.605
Effect size ++		<0.0001	<0.0001	<0.0001	<0.0001
Perineural invasion *					
Da	4 (6%)	-	-	-	
Ne	66 (94%)	-	-	-	

* Groups with N < 5 were excluded from analysis; ** Mann–Whitney test; *** Rank-biserial correlation; + ANOVA; ++ ω^2^.

**Table 5 biomedicines-13-02460-t005:** Relative expression of miR-185-5p, miR-141-5p, miR-21-5p, and miR-16-5p in colon cancer tissue according to disease progression.

Characteristics	N (%)	Relative Expression in Tumor Tissue (Median, Range)			
		miR-185-5p	miR-141-5p	miR-21-5p	miR-16-5p
Progression					
Yes	9 (15%)	1.158 (0.264–2.354)	0.055 (0.0002–0.298)	207.481 (13.687–533.130)	9.981 (0.028–55.814)
No	59 (82%)	0.800 (0.266–2.041)	0.038 (0.0002–1.069)	155.913 (13.594–729.997)	6.871 (0.053–95.904)
Unknown	2 (3%)	-	-	-	
*p* **		0.479	0.949	0.978	0.978
Effect size ***		0.149	0.015	0.008	−0.008
5-year survival					
Yes	54 (77%)	0.831 (0.266–2.041)	0.036 (0.0002–0.397)	149.500 (13.594–729.997)	6.832 (0.053–70.640)
No	15 (2%)	0.764 (0.264–2.354)	0.052 (0.0002–1.069)	184.642 (13.687–658.571)	8.889 (0.028–95.904)
Unknown	1 (1%)	-	-	-	-
*p* **		0.871	0.690	0.451	0.657
Effect size ***		0.029	0.069	0.130	0.077

** Mann–Whitney test; *** Rank-biserial correlation.

## Data Availability

The original contributions presented in this study are included in the article. Further inquiries can be directed to the corresponding author.

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
