# Peer review of "Diagnostic Relevance of miR-185, miR-141, and miR-21 in Colon Carcinoma: Insights into Tumor Sidedness and Reference Gene Selection"

_biomedicines, 2025, doi:10.3390/biomedicines13102460_

Round 1

Reviewer 1 Report

Comments and Suggestions for Authors

In this paper, the authors assessed the Clinical Relevance of miR-185, miR-141, and miR-21 in CRC and their association with several clinical outcomes. Despite the overall concept being interesting, this study contains several issues that need to be addressed:

Title:

The title should be revised to reflect the main aim and findings of the study, as these miRNAs were previously investigated.

Abstract:

It would be better to explain the rationale for selecting these specific miRNAs, highlighting what previous literature has reported about them, so that the aim of your work becomes clearer.

Introduction

The Introduction is long and should not have subtitles.

Methodology:

  • What are the normality tests that you used to differentiate between normally and non-normally distributed data?
  • While evaluating the stability of the housekeeping genes, what was the reference gene used? Kindly write it in more detail.
  • The rationale for exclusively using miR-151a-3p for normalization requires greater explanation.

Results

  • What is the value of writing ω² beside the P values?
  • Why did the authors use the 2^(-ΔCt) instead of delta delta Ct? Why didn’t they normalize the data relative to the control?’

Reviewer 2 Report

Comments and Suggestions for Authors

The manuscript addresses a clinically relevant and important topic - evaluating tissue-based microRNA signatures in colon cancer. It is well structured, the methodology is adequately described, and the results provide useful insights into miR-185, miR-141, and miR-21's diagnostic utility, as well as miR-16's unsuitability as a reference gene. The findings are consistent with previous reports, while also adding new insights regarding location-specific effects. The manuscript, however, needs clarification, expansion, and refinement before it can be published.

Comments:

  1. It would be helpful if the author highlighted the location-specific findings (right vs. left) in the abstract, as this is one of the study's more noteworthy contributions.
  2. The authors confirm that miR-185, miR-141, and miR-21 contribute to colon cancer. However, the novelty of the findings could be better emphasized. Nevertheless, their implications should be highlighted-how do these findings affect tissue miRNA biomarker applicability?
  3. The modest but significant sidedness effect observed for miR-185 requires further investigation. In light of the embryological, molecular, and microbiome differences between left- and right-sided colon cancers, what mechanisms might underlie this divergence? Could it be a result of differential Wnt/β-catenin signaling or EMT programs, where miR-185 plays a regulatory role?
  4. The cohort is limited to colon adenocarcinoma patients in the phase I-III stage. As a result, the data are internally consistent. This lack of survival associations may be explained by the absence of advanced or metastatic cases, which should be explicitly acknowledged by the authors.
  5. Most of the results are presented in tabular format, making it difficult to appreciate expression differences at a glance. For better understanding tumor vs. normal differences and localization effects, graphical visualizations (e.g., boxplots, violin plots or bar graphs) should be used in place of tables. As a result, readability would be greatly improved and best practices for biomarker studies would be followed.
  6. It would be more informative to include a fitted linear regression curve (with confidence intervals) in the correlation plots. In addition to visually conveying the strength and direction of the associations, this would help readers assess whether the correlations are consistent across the entire expression range rather than driven by outliers.
  7. A graphical model summarizing the main findings should be included by the authors. To better understand how these miRNAs contribute to colon cancer biology, a schematic incorporating tumor–normal differences, the role of each miRNA (miR-185, miR-141, miR-21, and miR-16), and the influence of tumor location would be helpful.
  8. Check the manuscript carefully for minor grammatical and typographical errors (e.g., "mir-14-5p" instead of "miR-141-5p" in Discussion).

Round 2

Reviewer 1 Report

Comments and Suggestions for Authors

I would like to thank the authors for considering my comments and for the revisions they made to the manuscript. The authors have provided a detailed point-by-point response and have made significant improvements to the manuscript.

Good luck.